# Underwater Drag Reduction Applications and Fabrication of Bio-Inspired Surfaces: A Review

**DOI:** 10.3390/biomimetics10070470

**Published:** 2025-07-17

**Authors:** Zaixiang Zheng, Xin Gu, Shengnan Yang, Yue Wang, Ying Zhang, Qingzhen Han, Pan Cao

**Affiliations:** 1School of Mechanical Engineering, Yangzhou University, Yangzhou 225009, China; 2Jiangsu Key Laboratory of Surface Strengthening and Functional Manufacturing, Yangzhou University, Yangzhou 225009, China; 3National Key Laboratory of Autonomous Marine Vehicle Technology, Harbin Engineering University, Harbin 150001, China

**Keywords:** bio-inspired drag reduction, non-smooth surface, superhydrophobic surface, modified coating, drag reduction mechanism

## Abstract

As an emerging energy-saving approach, bio-inspired drag reduction technology has become a key research direction for reducing energy consumption and greenhouse gas emissions. This study introduces the latest research progress on bio-inspired microstructured surfaces in the field of underwater drag reduction, focusing on analyzing the drag reduction mechanism, preparation process, and application effect of the three major technological paths; namely, bio-inspired non-smooth surfaces, bio-inspired superhydrophobic surfaces, and bio-inspired modified coatings. Bio-inspired non-smooth surfaces can significantly reduce the wall shear stress by regulating the flow characteristics of the turbulent boundary layer through microstructure design. Bio-inspired superhydrophobic surfaces form stable gas–liquid interfaces through the construction of micro-nanostructures and reduce frictional resistance by utilizing the slip boundary effect. Bio-inspired modified coatings, on the other hand, realize the synergistic function of drag reduction and antifouling through targeted chemical modification of materials and design of micro-nanostructures. Although these technologies have made significant progress in drag reduction performance, their engineering applications still face bottlenecks such as manufacturing process complexity, gas layer stability, and durability. Future research should focus on the analysis of drag reduction mechanisms and optimization of material properties under multi-physical field coupling conditions, the development of efficient and low-cost manufacturing processes, and the enhancement of surface stability and adaptability through dynamic self-healing coatings and smart response materials. It is hoped that the latest research status of bio-inspired drag reduction technology reviewed in this study provides a theoretical basis and technical reference for the sustainable development and energy-saving design of ships and underwater vehicles.

## 1. Introduction

Marine transportation is one of the most important modes of transport in international trade. Fuel consumption during ship movement is mainly used to overcome frictional resistance, accounting for 60–80% of the total resistance. At present, global warming and the energy crisis are the two major challenges facing mankind, and transportation is one of the main factors leading to these problems. Reducing the greenhouse gas emissions and energy consumption of ships has become the focus of research, among which drag reduction technology is an important research direction. Reducing the surface friction resistance by 10% can save approximately 3.75% energy, and reducing such friction is thus essential to reduce the energy consumption and improve the performance of ships and underwater vehicles. To improve energy efficiency and reduce costs, researchers have been exploring various bio-inspired drag reduction methods. Fish have developed efficient adaptations through 3.7 billion years of natural selection and evolution, including streamlined bodies and specialized skin textures that exhibit significant low-drag capabilities. The development of bio-inspired technologies has led to the trend of bio-inspired drag-reducing surfaces, which provide abundant inspiration for designing functional surfaces with drag-reducing properties [1,2]. Natural biological systems have evolved over centuries to develop unique functional surfaces or special physiological functions to adapt to complex environments, and these functional designs can provide inspiration for solving technological challenges. The main sources of the various types of bio-inspired drag reduction methods are shown in Figure 1. Bio-inspired studies have shown that ribbed features can reduce frictional drag by controlling the wall shape to modify the flow field in the boundary layer. 

Since the discovery of the drag-reducing ability, numerous control methods and techniques have been developed to reduce frictional resistance; however, most of these are active approaches, meaning that they require additional power inputs and can make the devices more complex [3]. Bio-inspired surfaces, as a passive drag reduction method, have received significant attention because they draw on the structure and function of natural biological surfaces [4]. The drag reduction rate (DR) and Reynolds number (Re) are two extremely important parameters for evaluation of the effectiveness of underwater drag reduction methods, and their synergistic effect allows for the quantitative characterization of the drag reduction performance. In particular, the drag reduction rate is used as a physical quantity to measure the extent to which a certain type of drag reduction measure reduces the resistance of a fluid during fluid flow, usually expressed as a percentage.

Drag Reduction (*DR*):(1)$$DR\%=\frac{F_{0}-F_{s}}{F_{0}}\times 100\%$$
where $F_{0}$ is the fluid resistance without drag reduction and $F_{s}$ is the fluid resistance after drag reduction.

Meanwhile, the Reynolds number is a dimensionless number that can be used to characterize the fluid flow, defined as the ratio of inertial force and viscous force. When the Reynolds number is small, the viscous force on the flow field is greater than the inertial force and the fluid flow is more stable, showing laminar flow. As the Reynolds number increases, the inertia force gradually dominates, the fluid flow becomes unstable, and turbulence occurs [5,6,7].

Reynolds number (*Re*):(2)$$Re=\frac{\rho VL}{\mu }$$
where *ρ* is the fluid density, *V* is the fluid velocity, *L* is the characteristic length, and *µ* is the fluid viscosity coefficient.

Research on functional surfaces based on biological prototypes has gradually developed into three major technological paths: bio-inspired non-smooth surfaces, bio-inspired superhydrophobic surfaces, and bio-inspired modified coatings. These technologies change the solid–liquid interface properties through different physical mechanisms, thus realizing effective regulation of the boundary layer flow. The skin of sharks and other fish are typical non-smooth surfaces, covered with micron-level ribbed dermal denticle structures that can effectively inhibit turbulence bursts through the generation of secondary vortices, thus reducing wall shear stress. Experiments have shown that the structure in a specific flow rate can be realized under the drag reduction effect of 8–15%.

In contrast, superhydrophobic surfaces inspired by plant effects, such as lotus leaves, can theoretically reduce frictional drag by 20–30% through the utilization of micro- and nanocomposite structures to lock the air layer and form air–liquid slip boundary conditions. Researchers have begun to explore synergistic drag reduction mechanisms in recent years, including combining the ribbed structure of shark skin with superhydrophobic properties, which was found to improve drag reduction by approximately 40% compared with a single structure.

Modified coating technology, on the other hand, maintains the functional stability of a surface through the development of low-surface-energy polymers or self-healing coatings from the perspective of material chemistry. Elastomeric materials such as polydimethylsiloxane (PDMS) can effectively adapt to the change in flow field at different flow rates due to their good deformation recovery ability. It is worth noting that the mucus layer secreted by the epidermis of marine organisms has provided inspiration for new biomimetic coatings. This type of biological mucus can cause dynamic drag reduction by changing the local viscosity, and its rheological properties open a new direction for the development of smart response coatings. Although the mechanisms of the three technical approaches are different, they all follow the principle of synergy between structure and function and complement each other in key aspects of fluid mechanics, such as reducing turbulence fluctuation and delaying flow separation. In practical applications, bio-inspired non-smooth surfaces adapt to the multi-phase flow environment, but their widespread adoption is limited by their processing cost; bio-inspired superhydrophobic surfaces provide significant drag reduction in microfluidics and other scenarios, but their durability is insufficient; modified coating surfaces have a wide range of applications but are greatly influenced by environmental factors. In the future, composite applications can be explored to take performance into account, intelligent adaptive surfaces can be developed to deal with variable flow fields, and the correlations between microscopic mechanisms and macroscopic effects can be deepened to further enhance the practical value and scientific contribution of the technology.

Herein, we systematically analyze the latest progress in bio-inspired drag reduction technology, focusing on analyzing the surface morphology parameters and exploring synergistic mechanisms under the multi-scale coupling effect. Comparing the applicable scenarios and technical bottlenecks of different methods provides a theoretical basis and technical reference for the energy-saving design of ships and underwater vehicles. The subsequent second to fourth sections of the article explore the three major technological paths in depth in sequence; in each section, the technological path and the corresponding bio-inspired objects are briefly introduced, the mainstream fabrication methods are summarized, the drag reduction effects and mechanisms are discussed, and, finally, the prospective engineering applications of the bio-inspired drag reduction technology are outlined.

## 2. Bio-Inspired Non-Smooth Surfaces

### 2.1. Inspiration and Morphology

In the public’s commonly held perception, the smoothness of the surface of an object and the magnitude of its frictional resistance are negatively correlated; that is, the smoother the surface, the lower its frictional resistance. However, the U.S. National Aeronautics and Space Administration (NASA), under the Langley Research Center, conducted related research that completely overturned this traditional concept. The results of this study clearly showed that surfaces with non-smooth groove structures can effectively reduce the frictional resistance of the wall [8,9]. For example, a three-dimensional staggered groove microstructure with a 90° orientation reduced the drag force by 11% compared with a smooth surface [10].

Compared with other drag reduction methods, non-smooth microstructures can manipulate the turbulent boundary layer and achieve stable drag reduction without additional energy input [11]. In nature, the surface structures of organisms have evolved over time and adapted to environmental conditions to exhibit significant drag reduction. For example, the unique biological feature of fish skin is the scales, which are embedded in a pliable dermis and covered by a viscoelastic mucus or pliable epidermal layer. This structure allows the entire fish skin system to act as a dynamic elastic energy-absorbing coating with microroughness, absorbing external turbulent pulsations and contributing to the reduction of frictional drag [12]. Analyzing the surface microstructure of fish scales and revealing their biological characteristics associated with skin friction resistance reduction can provide inspiration for the development of bio-inspired drag reduction technologies. 

The dolphin’s flexible skin reduces drag through elastic vibration absorption and is structured with transversely corrugated microgrooves that are capable of significantly affecting the underwater drag reduction rate by altering the flow velocity and drag distribution near the wall and increasing the thickness of the viscous bottom layer [1]. The skin of dolphins, along with the viscoelastic properties of the underlying fat layer, has been reported to produce ridge-like deformations at supercritical swimming speeds that are perpendicular to the body axis and flow direction. These deformations generate bound vortices, which help to delay the transition to turbulence and enhance boundary layer formation.

The morphology of the pufferfish exhibits a streamlined configuration that gradually widens from the mouth to the central region of the eye and then narrows from the mid-abdomen to the tail. The streamlined body surface of the pufferfish reduces frictional resistance by reducing differential pressure resistance, promoting laminar flow formation, delaying fluid separation, and reducing turbulence-induced vortices [2]. In addition, the surface of the pufferfish’s skin is covered with a scattering of small spines that are wrapped in an elastic outer skin, forming a unique system that absorbs turbulent energy. When the pufferfish is in a steady state, these small spines affect water flow, reducing the momentum transfer between the fluid and the solid film and lowering the fluid resistance during swimming [13].

Shark skin has gained attention for its smooth and clean properties, inspiring the design of numerous clinical and engineering products. Shark skin surfaces consist of many micrometer-scale grooves and ridges, with these structures varying in size on different parts of the shark’s body [14,15]. The discovery that non-smooth groove structures in shark skin have a positive impact on surface drag reduction was first made in the 1980s. Since then, scientists from different disciplines have studied microstructured surfaces inspired by a number of fast-swimming animals and investigated the mechanisms underlying the drag reduction ability of these microstructures [16,17]. The tiny denticles on shark skin play a decisive role in drag reduction and antifouling; however, due to their complex structure, they are difficult to fabricate on a large scale. Thus, scientists usually simplify them to ribbed structures with triangular, rectangular, and other cross-sections. Scientists have conducted in-depth research on shark skin and found that its non-smooth surface structure can greatly reduce resistance to the water current when the shark is swimming at high speed, giving excellent drag reduction performance. The shape of the ribs has a great influence on the drag reduction effect, especially when the ribs are parallel to the direction of flow [18,19,20]. In addition, ribs with sharp tips show optimal performance in reducing wall shear stress [21,22,23]. Surface textures based on sharkskin groove design can reduce wall frictional resistance by up to 10% [24]. This design has been used in aerospace and swimming; for example, the rib design on commercial airplanes can reduce drag by 2% and the sharkskin swimsuits of Olympic swimmers can reduce drag by 7% [25].

The tuna is known for its unique swimming performance, as well as being a highly nutritious and long-lived swimmer; it swims at speeds of more than 15 m per second and can reach a top speed of 160 km per hour. The tuna is a faster swimmer compared with sharks and dolphins, being capable of instantaneous speeds of up to 1.5 m per second. Despite the attention given to the nutritional value of tuna, the multi-layered structure and mechanical properties of its skin have not been fully investigated [26]. Tuna skin consists of five layers: the mucus layer, epidermis, scales, dermis, and collagen fibers. Its excellent hydrodynamic performance is attributed to the streamlined body and the unique structural and material properties of the skin, with the epidermal layer overlying hard fish scales, which are flexible and capable of storing mechanical energy during swimming, making swimming more efficient. In addition, there are distinct high- and low-velocity stripes on the surface of the tuna’s stacked tile-like scales, and these unique stripes reduce the velocity gradient along the normal direction, resulting in a drag-reducing effect [12,27,28,29].

Mudskippers are fish that can swim quickly in muddy environments and have a unique drag reduction mechanism. Mudskippers not only secrete a film of lubricating mucus under the compression and scraping of water and sediment, but their surface microstructure also retains the mucus well and prevents its rapid loss. In addition, the flexibility of the loach’s body maximizes the drag reduction rate through deformation, keeping the drag reduction rate consistently high over a wider range of flow rates [30,31].

Beluga whales and killer whales are both mammals with excellent swimming abilities. Beluga whales can swim fast in water due to the presence of wavy and regular ridges on the surface of their skin, which are distributed almost perpendicular to the direction of the water flow and have a sinusoidal cross-sectional shape. As a high-speed toothed whale, killer whales have a unique skin structure that is thought to have drag-reducing properties. It is worth noting that the ridges on the skin of killer whales also have regular shapes and obvious geometric features, with their distribution being almost perpendicular to the direction of the flow of water; additionally, their cross-sectional shapes are similar to sinusoidal waves, with smooth transitions between the peaks and troughs [32,33]. The mainstream bio-inspired objects are shown in Figure 2. 

Bio-inspired drag-reducing research relating to other fishes has also attracted much attention. For example, the pecten of the flounder is inclined, with millimeter-sized elliptical fins and sub-millimeter-sized spines; this double-layered structure reduces the drag on the flounder when it swims. The mantis shrimp, a ferocious predator in the ocean, has a body structure that makes it a perfect predatory machine. Its unique abdominal structure reduces the drag and vibration of the prey. The swordfish has micropores distributed on its skin that communicate with the external environment, and it has been hypothesized that swordfish can lubricate the surface of their skin by secreting oil to reduce frictional resistance. Bubbles have been observed on the skin of the sailfish, which may help to reduce frictional resistance by retaining air. Although these hypotheses aim to elucidate why fish can swim so fast, they have not been conclusively verified. Analyzing the surface microstructure of fish scales may provide key clues to the biological functions of fish scales in reducing drag, which could lead to the development of drag reduction technologies with practical applications [3,4,34].

### 2.2. Fabrication Methods

The fabrication methods of bio-inspired non-smooth grooved surfaces have been developed and diversified in recent years. The properties of non-smooth surfaces and superhydrophobic surfaces are derived from their microstructures, which are the key elements that result in the drag reduction effect. Theoretical explorations have been quite successful in the field of bio-inspired drag reduction, and many scholars have analyzed the drag reduction mechanism and constructed in-depth models; however, the real-world application of these mechanisms poses several problems. Due to the lack of mature, efficient, and suitable microstructure fabrication technologies, the process of developing bio-inspired non-smooth surfaces from theoretical conception to practical applications is difficult, making it difficult to fully utilize their potential value in many fields, including ship transportation [35]. The current mainstream fabrication methods include the hybrid method of sintering and coating [13], roller shaft hot embossing, micro-end-milling and micro-fly-milling [11,22], computer numerical control (CNC) machining [31], 3D printing technology [4,12,14,23,36], and laser ablating technology [28] (Table 1).

The hybrid approach of sintering and coating results in a superior preparation technique due to the combination of the two processes. In the sintering process, a porous metal substrate with a rough surface is fabricated via rapid sintering, which provides favorable conditions for the penetration of the coating into the substrate, thus significantly enhancing the wear resistance of the surface in practical applications. The observation and analysis of the droplet penetration phenomenon indicate the feasibility of the hybrid process. The final bio-inspired surface has a unique structure consisting of a conical protrusion array covered with an elastic layer, which we denote as a novel drag-reducing surface CPES [13]. The roller shaft hot embossing process has led to the development of a new roller thermal imprinting machine designed for the efficient production of linear grooved films. By heating the mold rollers and precisely regulating the pressure, temperature, and speed, the technology enables the polymer film surface to replicate the microstructure of the mold surface. The mold rollers act as the key carriers in this process, and the synergistic action of heat and pressure causes the polymer film to plastically deform into the desired microstructure. Precise preparation of the curved grooves can be realized with the help of micro-milling technology. Plexiglass flats are used as the machining material and a 60° double-edged, flat-bottomed cutter is used for machining. The optimization of the groove machining effect is achieved by precisely controlling key parameters such as spindle speed, tool feed rate, and cutting step. During this process, the parameters interact with each other to determine the accuracy and quality of the final groove. The surface produced by the micro-fly-milling technique has been shown to be superior in terms of the formation of shape accuracy and surface roughness, compared with the micro-end-milling technique, and was therefore identified as an ideal strategy for subsequent work [11,22]. Computer numerical control (CNC) machining technology can transform aluminum substrates into bio-inspired non-smooth grooved surfaces with specific microstructures by engraving directly on the substrate. Its significant advantage lies in its high processing accuracy, allowing accurate control of the size and shape of the microstructure of the non-smooth surface, perfectly meet the processing needs of various complex non-smooth surfaces, accurately control the surface roughness and texture, and ensure that the parameters of the processed surfaces meet extremely accurate standards, thus laying a solid foundation for the improvement of product quality and performance [31]. Three-dimensional printing technology can create textured surfaces that exhibit specific scale orientations by manipulating 3D printers to accurately mimic the surface characteristics of fish skin [4,12,14,23,36]. It is worth noting that, compared with smooth surfaces, the tooth surface obtained using 3D printing technology shows unique advantages under specific conditions, which can effectively reduce energy consumption and result in more ideal energy utilization in related applications [35]. Laser etching technology uses a template of a specified material as a substrate and a femtosecond laser to create bio-inspired groove surfaces based on the designed curves. This technology has high processing accuracy and can realize precision processing at the micro-nanometer level; in addition, it is a non-contact processing method. Furthermore, the developed material is widely adaptable, allowing it to withstand physical damage; additionally, the laser parameters can be adjusted to meet the processing needs of different materials [28,37,38].

### 2.3. Drag Reduction Effect and Mechanisms Underlying Non-Smooth Surfaces

Smooth surfaces struggle to effectively handle complex flow behaviors near the wall in turbulent environments, as illustrated in Figure 3a, while bio-inspired non-smooth surfaces show significant drag reduction advantages by virtue of their unique structure, as shown in Figure 3b. Research on the drag reduction technology of bio-inspired non-smooth groove surfaces has resulted in numerous findings. At present, most researchers use a combination of numerical simulations and experimental methods, taking full advantage of both methods to identify the mechanisms underlying drag reduction and to optimize surface designs. Numerical simulations can accurately analyze complex flow fields, efficiently explore multiple parameter combinations, and reduce research costs and time, while experimentation provides reliable data in real-world environments, allowing for verification of the accuracy of the simulation results. Using this complementary approach, researchers can better understand the drag reduction mechanisms and optimize surface designs to enhance the drag reduction effect and provide strong support for practical applications.

In terms of microstructural design, researchers have explored the drag reduction effects of various bio-inspired surfaces using different fabrication techniques. The drag reduction effects of different bio-inspired non-smooth surfaces are shown in Figure 4. DAI et al. fabricated shark skin-like textured surfaces with different scale orientations using 3D printing and found that the viscous drag of surfaces with 90-degree scale orientations was reduced by 9% [14]. Wu et al. prepared five microstructured surfaces using micro-fly-milling and found that rectangular grooves had the best drag reduction, with a maximum drag reduction of 9.70%. The averaged drag reduction rate for case fluid velocities in the range 0.5–4.5 m/s is 13.05% [11]. Gu et al. designed wing profiles based on the structure of the abdominal segments of the mantis shrimp and found that the drag reduction rate reached 15.33% when the flow velocity was 10 m/s and the number of wing profiles was 5 [34].

The analysis of drag reduction mechanisms shows that the bio-inspired surface effectively reduces the surface friction by changing the interaction mode between the fluid and the wall. Ao et al. investigated the drag reduction mechanism underlying three-dimensional spherical coronal microstructures and found that the drag reduction rate of the negatively oriented protruding spherical coronal microstructures was as high as 24.8% [21]. Using numerical simulations, Zhang et al. found that the drag reduction efficiency of V-shaped and serrated microstructures could reach 8.76% and 10%, respectively, under specific conditions [39]. Feng et al. investigated the drag reduction performance of conical protruding structures and elastic surfaces in laminar and turbulent flows, respectively, and found that the drag reduction performance of CPES bio-inspired surfaces in turbulence is 11.5–17.5% [13]. Fan et al. investigated the flow of a bio-inspired sharkskin ribbed surface in a fully turbulent pipe using the large eddy simulation (LES) method and found that the rib reduced drag by 21.45% at a Reynolds number Re of 40,459 [40,41].

**Figure 4 biomimetics-10-00470-f004:**
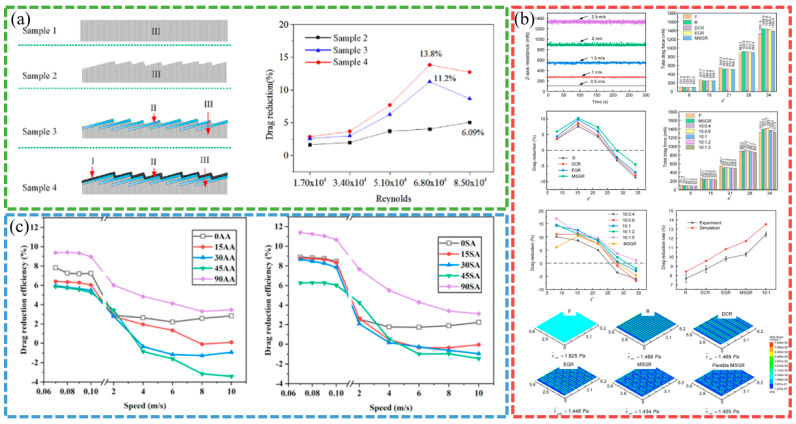
Drag reduction effect of different bio-inspired non-smooth surface [12,23,42]. (**a**) Drag reduction results of the bio-inspired surfaces compared with a smooth surface (Sample 1). (**b**) DR effects on different surfaces. (**c**) Plot of modeled drag reduction efficiencies for different riblet surfaces: aligned arrangement and staggered arrangement.

Significant progress has also been made in the optimization and application of bio-inspired surfaces. He et al. fabricated a bio-inspired flounder bilayer-structured surface (BFTSS) using 3D printing with a drag reduction rate of 19% in forward flow and 4.2% in reverse flow [4]. Zheng et al. designed a transverse bio-inspired groove surface with a frictional drag reduction ratio in the velocity range of 2~12 m/s, reaching 26.91% [33]. Using experimental and numerical simulations, Cui et al. showed that the maximum drag reduction rate of flexible multi-stage gradient ribs (MSGRs) could reach 16.8% at a flow velocity of 0.5 m/s [23]. Mawignon et al. conducted a simulation using the shear stress transport k-omega turbulence model and found that the ribs effectively reduce drag in different flow regimes. For laminar and turbulent flow, the drag reduction efficiency of ribs decreases and then increases with the increase in rib orientation angle, and reaches optimal drag reduction at 90° orientation angle; additionally, the staggered rib surfaces are more effective than the neatly arranged ones, and the drag reduction rates of 90° staggered rib surfaces in laminar and turbulent flow states are approximately 11.3% and 6%, respectively [42].

The high- and low-velocity stripes formed on the bio-inspired tile-like fish scale surface are closely associated with the fluid motion. The effects of different factors on the drag reduction rate are shown in Figure 5. Using numerical simulations, Chen et al. found that the maximum drag reduction rate of the bio-inspired dual-composite surface reaches 25.7%, attributed to the vortex effect of the flexible skin. The significance of these stripes increases as the spacing between adjacent fish scales and the tilt angle of the fish scales increase; however, the velocity amplitude of the stripes decreases as the exposed length of the fish scales increases. The bio-inspired fish scale surface reduces the gradient velocity which, in turn, reduces the wall shear stress, leading to drag reduction. The total drag force on the bio-inspired fish scale surface was measured in a recirculating tank, and the maximum drag reduction was 10.26% at a Reynolds number of 39,532. The drag reduction performance decreased as the fish scale spacing increased. An analysis of the drag reduction mechanism using the CFD method found that longitudinal vortices are generated at the valley bottom of the bio-inspired fish scales, which can change the sliding friction into rolling friction, thus effectively realizing the drag reduction effect [26,28,29]. Hu et al. conducted numerical simulations based on the ridged features of beluga whale skin using the SST k-ω model and compared the viscous resistance and pressure drop resistance of the fitted structure at different widths and depths, as well as changes in pressure drop resistance. The fitted structure demonstrated the best drag reduction performance, and the fitted structure with a width of 30 mm and a depth of 0.7 mm achieved the best drag reduction effect of 4.18% [32].

Although research on bio-inspired non-smooth surfaces in the field of drag reduction has generated diverse design ideas and significant drag reduction effects, many challenges remain to be addressed. From microstructure design to the in-depth analysis of the hydrodynamic mechanism, the drag reduction performance of bio-inspired surfaces is affected by a combination of factors, including the fluid state and the geometric parameters of the surface; thus, it is necessary to further optimize the microstructure design, combine multi-disciplinary approaches to understand the mechanism underlying drag reduction, and verify its applicability under complex flow conditions using a combination of experiments and numerical simulations. In addition, balancing drag reduction efficiency and manufacturing costs in practical engineering applications remains an important issue for the industrialization of bio-inspired non-smooth surface technology.

## 3. Bio-Inspired Superhydrophobic Surfaces

### 3.1. Inspiration and Morphology

Bio-inspired non-smooth surfaces provide an important theoretical and experimental basis for drag reduction techniques by regulating the flow properties of turbulent boundary layers through microstructure design. In addition to the direct modulation of flow properties through microstructures, another effective drag reduction method is utilizing the gas–liquid interfacial slip effect of superhydrophobic surfaces (SHSs), which significantly reduces the fluid drag by changing the physical properties of the solid–liquid interface. Superhydrophobic surfaces are special surfaces with a water contact angle of more than 150° and a sliding angle of less than 10°. Their unique water repulsion properties have led to a wide range of applications in several fields [43,44]. These surfaces are biologically inspired by nature and significantly reduce the contact area between the fluid and the solid surface by forming a stable gas–liquid interface underwater, thus reducing the frictional drag [45,46,47]. Their drag reduction properties have shown great potential for use in underwater vehicles, pipeline transportation, droplet manipulation, and ocean engineering. In addition, superhydrophobic surfaces have attracted significant attention in cross-disciplinary applications such as fog collection, oil–water separation, and drug delivery, making them a hotspot in multi-functional materials research [48,49,50,51].

Superhydrophobic surfaces can significantly reduce viscous drag by forming an air layer between the solid wall and the liquid; the stability of this air layer is critical for realizing the drag reduction effect. By injecting air into the boundary layer or using the superhydrophobic surface to form an air layer, the no-slip boundary at the solid surface can be transformed into a slip boundary at the gas–liquid interface. This reduces the shear force in the boundary layer, delays the transition of the flow state, increases the thickness of the laminar boundary layer, and realizes the drag reduction effect. Superhydrophobic surfaces have significant advantages in reducing frictional resistance in liquid transportation, especially in the marine field, and their low frictional resistance can significantly reduce the drag during turbulent flow. Their drag reduction effect is affected by the surface structure and the Reynolds number [52,53,54,55]. Many organisms in nature have developed unique surface properties to adapt to complex environments through long-term evolution; these properties provide a rich source of inspiration for the design of superhydrophobic surfaces. The mainstream biomimetic objects are shown in Figure 6.

The lotus leaf is known for its unique superhydrophobic and self-cleaning ability—a property that stems from the synergistic action of the microstructures and waxy layers on its surface. The surface of the lotus leaf is covered with many micron-sized papillae, which have a diameter of approximately 2–8 microns and a height of approximately 6–12 microns. These papillae structures form a hierarchical surface at the micro- and nanoscale. This microstructure allows the surface of the lotus leaf to form a layer of air when in contact with a water droplet, thus reducing the actual contact area between the water and the surface. The surface of the lotus leaf is also covered with a layer of wax, which has low-surface-energy properties and further enhances its hydrophobicity. Due to the microstructure of the surface and the low surface energy of the wax, when a water droplet falls on the surface of a lotus leaf, it forms a nearly spherical shape and rolls easily. As they roll, the water droplets carry dust and dirt from the surface, resulting in the self-cleaning effect. This phenomenon is known as the lotus effect [56,57].

**Figure 6 biomimetics-10-00470-f006:**
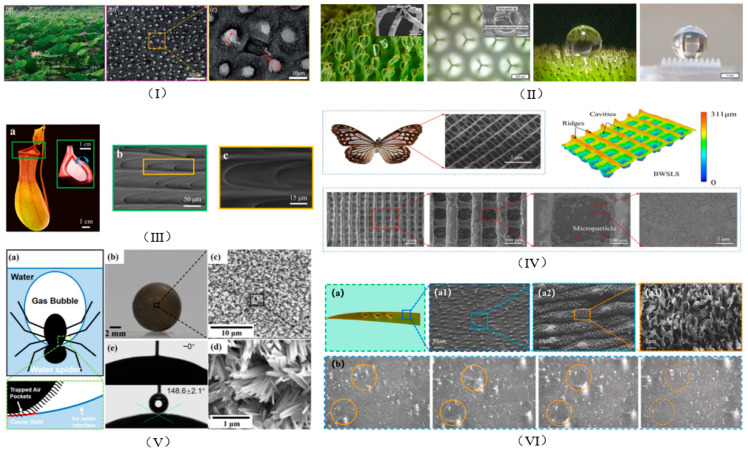
Microstructural features of the surfaces of mainstream bio-inspired objects: (**I**) SEM image of a lotus leaf and its papillae microstructure [56]. (**II**) Hairs and hierarchical structures inspired by them on a sagebrush [58]. (**III**) SEM image of a mouth of a hogwash plant with many arched microcavities [59]. (**IV**) SEM image and super depth-of-field microscopy image of a butterfly wing scale structure of a Pallanikameranus butterfly [60]. (**V**) Schematic of the bubble trapping process in water spiders and optical and SEM images of the surface of a modified superhydrophobic sphere [61]. (**VI**) SEM image of an iris leaf and the phenomenon of condensation of droplets on the iris leaf [62].

Salvinia molesta is a floating fern with a distinctive agitator-shaped hierarchical hair structure on the surface of its fronds. Its surface structure can maintain a stable, thick layer of air underwater, a property known as the “salvinia effect.” When a sage blade is submerged in water, its surface can maintain a stable, thick air layer; this is due to the structure and elasticity of the microhairs, which resist water fluctuations, thus prolonging the retention time of the air layer. This stable air layer creates an air lubrication layer underwater, reducing the contact area between the water and the blade surface, thus significantly reducing hydrodynamic drag reduction (HDR) [58].

The butterfly Parantica melaneus lives in tropical and subtropical regions. Its wing scales have a unique microstructure consisting of parallel longitudinal ridges and regularly arranged rectangular cavities. This microstructure can form a stable gas film on its surface. When the butterfly’s wings come into contact with water, the water forms droplets under the action of the gas film and slides down quickly, keeping the wings dry. Therefore, this structure gives it excellent superhydrophobicity, enabling it to maintain flight in humid tropical environments. It not only effectively reduces surface energy and water droplet adhesion but also maintains a stable air–liquid interface in the rain, thus reducing flight resistance [59].

Hogweed is a unique insectivorous plant whose insect-trapping mechanism relies heavily on its unique lip structure. The lip surface of Hogweed has a number of arched microcavities with sharp edges and a certain angle of inclination. When a liquid drop falls on the lip surface, these structures direct the liquid unidirectionally to the outer edge of the cage and can even overcome gravity to hold the liquid on the curved surface. The sharp edges and arching curves of the lip of the hogweed work together to achieve superb liquid repulsion and form a stable contact angle between the liquid and the solid surface, resulting in superhydrophobicity [60].

Water spiders are semi-aquatic spiders that can live and hunt underwater. Their abdomens are covered with a large number of microscopic hairs, with a layer of low-surface-energy wax on the surface of these hairs. This unique structure and chemistry make their abdomens superhydrophobic. They utilize this superhydrophobic property to trap and carry a large air bubble underwater, known as a “physical gill,” which not only provides the spider with oxygen but also significantly reduces its resistance underwater, allowing it to move more efficiently [61].

The leaves of the Iridaceae plant family have unique microstructures that allow them to adapt to high-humidity environments and exhibit excellent superhydrophobic properties. The special microstructure of the surface of the leaf gives it a very high static contact angle (CA), which can reach approximately 165°, and a rolling angle (RA) of only 2°, which means that water droplets can roll off the surface of the leaf quickly, thus exhibiting excellent low adhesion [62]. This superhydrophobic property allows the leaves to drain quickly in high-humidity environments, thus preventing excessive water accumulation.

The surface structures of many natural organisms also provide important insights into the study of bio-inspired superhydrophobic surfaces. For example, the periodic micro-nanostructure of spider silk, which can form water droplets and transport them in a directional manner under high humidity, provides a unique template for the design of superhydrophobic surfaces. The micro-nanostructure on the back of a desert beetle collects water from mist due to the clever distribution of its superhydrophobic and hydrophilic regions. These natural phenomena suggest that there is a close relationship between the micro- and nanostructures of biological surfaces and their specialized functions. Although the understanding of the relationship between the structure and function of these biological surfaces is still at an exploratory stage, an in-depth study of the surface microstructures of these natural organisms will undoubtedly provide key clues to the biological mechanism underlying their superhydrophobicity, which will in turn lead to the development of new superhydrophobic materials with practical applications.

### 3.2. Fabrication Methods

Superhydrophobic surfaces have become a research hotspot in the fields of materials science and fluid dynamics in recent years due to their unique drag reduction properties and prospects for a wide range of applications, including self-cleaning and anti-icing applications [63,64,65,66,67,68,69,70,71]. There are various methods for preparing superhydrophobic surfaces. Traditional fabrication methods mainly involve regulating the surface roughness, chemical modification, or a combination of both, while current mainstream fabrication methods include 3D printing [48,59,60], spray coating [49], laser processing technology [37,55,72,73], photolithography [56,58], wet etching technology [61], electrostatic flocking technology [74], and electrochemical etching and anodization [66]. These methods impart superhydrophobic properties to the surface by building micro- and nanostructures and undergoing material chemical coating processes, such as coating. They also exhibit significant drag reduction under different flow rates and Reynolds numbers, as shown in Table 2.

Fabrication methods based on 3D printing technology process surfaces with complex microstructures by curing photosensitive resins layer by layer or using metal powders, endowing the surfaces with superhydrophobicity through chemical modification [59]. Laser processing techniques, on the other hand, further enhance the hydrophobicity of surfaces by etching periodic or irregular micro-nanostructures on metal or polymer substrates in combination with hydrophobic coatings [62]. Wet etching and photolithography are also widely used for the preparation of superhydrophobic surfaces by generating microstructured or nanowire arrays on the surface of the substrate through chemical reactions, followed by surface modification to achieve superhydrophobic properties [58,61]. Spray-coating technology has become an important surface preparation method due to its simplicity and efficiency, and superhydrophobic properties can be rapidly realized on a wide range of materials by uniformly spraying hydrophobic nanoparticles or polymer solutions onto the substrate surface [49]. Electrostatic flocking techniques combined with surface modification processes have also been used to prepare superhydrophobic surfaces, further enhancing the hydrophobicity and drag reduction properties of the surfaces through the synergistic effect of vertically fixed fiber structures and hydrophobic coatings [74].

The diversity and flexibility of these fabrication methods provide a wide scope for the practical application of superhydrophobic surfaces. Studies have shown that, by optimizing the design of micro- or nanostructures and surface chemical modification, superhydrophobic surfaces can exhibit significant advantages in underwater drag reduction, gas–liquid interfacial stabilization, and antifouling performance. With the further development of preparation technologies, superhydrophobic surfaces are expected to result in breakthrough applications in more fields in the future.

### 3.3. Drag Reduction Effect and Mechanisms Underlying Superhydrophobic Surfaces

Superhydrophobic surfaces (SHSs) show remarkable potential for application in drag reduction due to their unique micro- and nanostructures and low-surface-energy properties. In addition, the unique properties of the superhydrophobic grooved surfaces (SHGSs) resulting from the grooved structure give the surfaces superhydrophobic properties and can also regulate the behavior of the flow of liquids through the geometric parameters of the groove, which has important research value in microstructure flow field analysis. In recent years, researchers have investigated the drag reduction performance of superhydrophobic surfaces under different fluid environments and Reynolds numbers via various experimental and simulation methods. The drag reduction principles of SHS and SHGS are shown in Figure 7. The bio-inspired superhydrophobic surface has an elliptical shape and the center of gravity is shifted forward when the water droplets are rolling; additionally, its forward and backward contact angles are smaller than the static contact angle, as shown in Figure 8. This indicates that the microstructure reduces the contact area between the droplet and the surface, lowering the rolling resistance and accelerating the rolling speed of the droplet.

The drag reduction effect of superhydrophobic surfaces is closely correlated with their surface microstructure and the stability of the gas–liquid interface. Minsu Kim et al. used a rotational rheometer to show that the effective slip length of superhydrophobic surfaces in glycerol solution was approximately 137 μm; this can result in excellent drag reduction effect, showing that superhydrophobic surfaces significantly reduce the frictional resistance of the fluids through enhancement of the slip effect of the gas–liquid interface [58]. Similarly, Yangmin Chen et al. tested the underwater drag reduction performance of a bio-inspired butterfly wing microstructured surface (BWSLS) at moderate Reynolds number and found that the drag reduction rate could reach up to 20.56%, which is 6.4% higher than that of a monostructured surface [60]. These studies indicate that the micro- and nanostructures of superhydrophobic surfaces can effectively trap air bubbles and form a stable gas–liquid interface, thus significantly reducing the fluid drag.

The drag reduction effect of superhydrophobic surfaces in different fluid environments and Reynolds number conditions also showed significant differences, as shown in Figure 9. Using rheometer experiments, Wen Zhou et al. found that superhydrophobic surfaces with periodic rhombic arrays and irregular nanosheets had a drag reduction of close to 40% in fluids [62]. Changzhuang Yao et al. recorded the movement of superhydrophobic spheres in water using a high-speed video camera. They found that the air cavity formed by the superhydrophobic sphere significantly reduced the drag coefficient, and the drag reduction efficiency could reach over 90% [61]. These results demonstrate that the drag reduction effect of superhydrophobic surfaces not only depends on the design of the surface microstructure but is also closely correlated with the fluid flow rate, Reynolds number, and the stability of the gas–liquid interface.

The drag reduction effect of superhydrophobic surfaces still faces some challenges in practical applications. Jiangpeng Qu et al. used a homemade flat plate water impact testing device to show that the drag reduction rate of superhydrophobic coated surfaces with grooved structures exceeded 66.57% at Reynolds numbers ranging from 2 × 10^5^ to 5 × 10^5^; however, the drag reduction effect can gradually decrease at high Reynolds numbers [49]. In addition, Linsheng Zhang et al. found, through in situ monitoring, that the air layer of superhydrophobic surfaces in turbulent flow was gradually lost, leading to a decrease in the drag reduction effect [54,75]. These studies indicate that improving the stability and durability of superhydrophobic surfaces remains an important direction for future research.

In addition, as an important phenomenon of bionic drag reduction, the core principle of supercavitation is to transform the solid–liquid interface into a solid–air interface through the surface gas film to reduce drag. There is a common interface control logic with the drag reduction mechanism underlying superhydrophobic surfaces. The ability of the superhydrophobic micro- and nanostructures to capture bubbles can assist in stabilizing the supercavitating gas film, and the gas film formed via supercavitation can enhance the drag reduction effect of the superhydrophobic surface. At the technical level, the combination of supercavitation and the superhydrophobic surface has become a research hotspot: the low-surface-energy characteristics of superhydrophobic surfaces can reduce the attachment threshold of cavitation bubbles and promote the formation of gas films; at the same time, the low-surface-energy characteristics of superhydrophobic surfaces can reduce the adhesion threshold of cavitation bubbles and promote the formation of gas films. The microstructure can anchor the boundary of the gas film, inhibit the collapse of the cavitation bubble, and reduce the instability problem associated with traditional supercavitation. This synergistic system not only retains the stable drag reduction advantage of the superhydrophobic surface in medium- and low-speed scenarios but also overcomes the drag reduction bottleneck at high flow rates through the cavitation effect. An important direction for superhydrophobic surface drag reduction technology involves expanding to include more diverse working conditions.

In summary, there has been significant progress in the research on bio-inspired superhydrophobic surfaces in the field of drag reduction, with the drag reduction effects being verified in a variety of fluid environments and Reynolds number conditions. However, their practical applications still need to be optimized, particularly in terms of improving the stability of gas–liquid interfaces and surface durability. Future research should focus on structural design, material selection, and standardization of test conditions for bio-inspired superhydrophobic surfaces to promote their widespread application in drag reduction technology.

## 4. Bio-Inspired Modified Coatings

### 4.1. Modified Materials

On the basis of exploring bio-inspired non-smooth surfaces and bio-inspired superhydrophobic surfaces, remarkable progress has been made in integrating these two types of surfaces into composite superhydrophobic grooved surfaces, resulting in a promising approach for the development of bio-inspired superhydrophobic surfaces. The synergistic advantages of these two structures can be fully utilized. However, despite the significant potential of bio-inspired superhydrophobic surfaces in the field of drag reduction, challenges remain in terms of the stability and durability of the gas layer, especially in complex fluid environments. Through continuous research on the characteristics of non-smooth and superhydrophobic surfaces and to further enhance the drag reduction performance and realize the multi-functional synergistic effect, researchers have begun to explore the emerging biomimetic modified coating technology based on the design of micro- and nanostructures and chemical modification of the materials, as shown in Figure 10. The development of modified coated surfaces is one of the current research hotspots in the field of materials science and engineering; it focuses on optimization of the physical, chemical, and mechanical properties of coatings through the introduction of specific modified materials [76,77,78]. In recent years, researchers have extensively explored the use of various materials in coatings, including polymers, nanoparticles, metal oxides, and composites. The choice and combination of these materials not only determine the surface properties of the coating, but also directly affect its application in the fields of drag reduction, antifouling, and anticorrosion [79,80,81,82].

Polymer-based materials are widely used in the design of modified coatings due to their excellent flexibility and processability [83,84]. Polydimethylsiloxane (PDMS) is an ideal base material due to its low surface energy and good elastic modulus [85,86,87,88]. Through combination with other materials, such as hydrophobic silica [89] and zinc oxide [90], PDMS coatings can significantly enhance their superhydrophobicity and mechanical stability. Polymers such as waterborne polyurethane (WPU) and epoxy resin (ER) have also been extensively studied due to their wear resistance and chemical stability [86,91,92,93]. Through introducing fluorinated alkyl or siloxane modifications, the surface energy of these polymers can be further reduced to achieve superhydrophobic properties; thus, the multi-functionality of polymer coatings has shown promising applications in the fields of antifouling, drag reduction, and self-cleaning [94,95].

Inorganic nanoparticles are another important class of materials in modified coatings. Silicon dioxide (SiO_2_) [96,97,98], zinc oxide (ZnO) [99,100], and alumina (Al_2_O_3_) [101,102] nanoparticles have attracted significant attention due to their unique optical, electrical, and mechanical properties. These particles can significantly enhance the roughness and hydrophobicity of coatings by developing micro- and nanostructures. SiO_2_ nanoparticles form a homogeneous microstructure in coatings that helps to trap the air layer and reduces the contact area between liquids and the surface [103,104]. ZnO particles, on the other hand, have shown significant advantages in antifouling and drag-reducing coatings due to their photocatalytic and antimicrobial properties [105]. In addition, graphene and its derivatives have been widely used in the modification of coatings to enhance their abrasion and corrosion resistance, due to their excellent mechanical strength and electrical conductivity [106].

The design of composites offers wider possibilities for modified coatings. Through combining polymers with inorganic nanoparticles, synergistic effects can be achieved to optimize the comprehensive performance of the coatings [107,108]. The composite coating consisting of zeolite imidazole framework (ZIF-67) and polyurethane exhibits excellent superhydrophobicity and good anticorrosive properties [109]. Similarly, polysaccharide-based biomaterials are strong candidates for novel antifouling coatings due to their environmental friendliness and biocompatibility [110,111]. These materials can enhance the mechanical stability and durability of coatings while maintaining low surface energy through synergistic interactions with the polymer matrix.

Optimal selection of the material and design of modified coating surfaces is the key to realizing their multi-functionality. A reasonable combination of polymers, inorganic nanoparticles, and composites can meet the application requirements of coatings in the fields of drag reduction, antifouling, and anticorrosion and guarantee their long-term stability in complex environments [112]. Future research is expected to explore the development of novel materials and their application in coatings to promote the development of modified coating technology for higher performance and wider application.

### 4.2. Fabrication Methods

The core objective of preparing bio-inspired modified coated surfaces is to develop coating materials with excellent performance by modulating the microstructure and chemical composition of the surfaces, resulting in multi-functional properties such as drag reduction and antifouling. At present, the mainstream fabrication methods include chemical etching and surface modification techniques [94,113], spray deposition method [89,104], one-step spraying technique [97,114,115], surface-initiated atom transfer radical polymerization (SI-ATRP) [85], anodizing and dip-coating [96], single-crystal lithography combined with chemical particle modification [109], the Sol–Gel method [116], femtosecond laser ablation followed by chemical modification [117,118], and low-pressure plasma treatments [119]. As shown in Figure 11, these techniques can significantly improve the surface properties of coatings by building micro- and nanostructures, introducing low-surface-energy materials, or combining them.

The chemical etching and surface modification technique, as a classical method, provides an ideal template for the subsequent modification of low-surface-energy materials by forming rough micro- or nanostructures on the surface of the substrate. This method is easy to operate, inexpensive, and particularly suitable for the pretreatment of metallic substrates [113]. The spray deposition method results in layer-by-layer construction of coatings by uniformly dispersing functional nanoparticles in a polymer solution and utilizing the rapid evaporation of solvents during the spraying process, which has the advantage of accurately controlling the distribution of the thickness and roughness of the coatings [89]. The one-step spraying technique further simplifies the process flow and achieves simultaneous completion of the micro- and nanostructures and chemical modification through a single spraying, which significantly improves the efficiency of coating preparation [97,114].

The surface-initiated atom transfer radical polymerization (SI-ATRP) technique achieves precise modulation of the chemical composition of the surface by growing polymer brushes in situ on the surface of the substrate. This method not only avoids the delamination problem of conventional coatings but also allows for gradient regulation of surface energy by adjusting the density and length of polymer brushes [85]. The combination of anodic oxidation and dip-coating uses the anodic oxidation process to generate ordered nanopore structures on the metal surface, followed by dip-coating of functional materials to achieve multi-functionalization of surface properties, with the advantage of large-scale preparation of coatings with highly ordered structures [96].

Single-crystal lithography combined with chemical particle modification can replicate microstructural features of complex biological surfaces on smooth substrates through precise patterning at the micro- and nanoscales. This method is particularly suitable for the preparation of biomimetic coatings that require high-precision structural replication [109]. The Sol–Gel method, on the other hand, forms a uniform nanoporous network structure on the surface of the substrate through hydrolysis and condensation of the chemical solution, which has the advantage of generating uniform coating coverage from the microscopic to the macroscopic scales [116]. Femtosecond laser combined with surface treatment technology generates periodic nanostructures on the material surface using ultrashort pulsed laser light and subsequently imparts ultra-low rolling angle properties to the surface via chemical modification. This method can achieve extreme hydrophobicity of the surface without destroying the properties of the substrate [117,118]. Low-pressure plasma treatment technique, on the other hand, is particularly suitable for the surface functionalization of polymer substrates through the chemical interaction of reactive radicals with surface molecules to achieve a significant reduction in surface energy while maintaining the mechanical properties of the substrate [119].

Each of these fabrication methods has its own focus, but all of them have the core objective of regulating the surface microstructure and chemical composition. Chemical etching and anodizing focus on building rough physical structures, spray deposition and one-step spraying emphasize process simplicity and scalability, while SI-ATRP and single crystal lithography focus on the precise tuning of chemical composition. Sol–Gel and femtosecond laser technologies synergistically optimize performance through multi-scale structural design, while low-pressure plasma treatments achieve surface functionalization while maintaining substrate performance. The common point of these methods is that the multi-functional properties of the coatings are significantly enhanced through the synergistic effect of micro- and nanostructures and chemical compositions, which lays a solid foundation for the practical application of biomimetic modified coatings. In practical application, the selection of different fabrication methods requires taking the characteristics of the substrate, functional requirements, and process conditions into account. Chemical etching and anodic oxidation are suitable for the pretreatment of metal substrates, while spray deposition and one-step spraying are more suitable for large-scale industrialized production, and SI-ATRP and single-crystal lithography are mainly used in the bio-inspired field that requires the high precision of structure replication due to their high-precision characteristics. Sol–Gel and femtosecond laser technologies show unique advantages in the functionalization of complex surfaces and microscopic regions through multi-scale structural design. Low-pressure plasma treatment has become the preferred solution for surface modification of polymers and composites, due to its non-destructive effect on the substrate.

The current methods for preparing bio-inspired modified coating surfaces are developing in the direction of diversification and compositing. From the traditional chemical etching and spraying techniques to emerging technologies such as surface-initiated atom transfer radical polymerization (SI-ATRP) and low-pressure plasma processing, researchers are continuously exploring new preparation strategies to meet the high demands for coating performance in different application scenarios. With the further development of materials science and manufacturing technologies in the future, it is expected that combining different advanced preparation techniques will significantly improve surface energy modulation accuracy while maintaining coating uniformity. The continuous innovation and integration of these technologies will further expand the potential of using bio-inspired modified coatings in high-end applications in aerospace, marine engineering, and biomedical fields.

### 4.3. Drag Reduction Effect and Mechanisms Underlying Modified Coating

Modified coatings with superhydrophobicity prepared on non-smooth surfaces show significant potential for applications in drag reduction due to their unique surface grooves and wettability. It has been shown that by building micro- and nanostructures and reducing surface energy, such modified coatings can form a stable gas layer at the solid–liquid interface, effectively reducing fluid drag. Modified coatings based on polydimethylsiloxane (PDMS) and hydrophobic silica nanoparticles maintained a stable Cassie–Baxter state at a depth of 5 m underwater, with a maximum drag reduction of 25% and a slip length of approximately 70 µm [89]. The alumina-reinforced PDMS coating showed good drag reduction performance under low-Reynolds-number flow conditions, and the drag reduction rate was inversely proportional to the elastic modulus of the coating, suggesting that the mechanical properties of the coating have an important influence on the drag reduction effect. The drag reduction effects of micro-alumina- and nano-alumina-modified coatings are shown in Figure 12 [101]. The study’s results demonstrated that the bio-inspired modified coatings can significantly reduce the wall shear stresses and reduce the energy required to push the fluid or move the fluid through the channel by changing the chemical composition and topology of the surface.

Bio-inspired design has also seen significant progress in the study of drag-reducing coatings. A non-smooth surface inspired by the shield scale structure of shark skin was constructed as micro- and nanostructures using a simplified design and combined with the modification of zeolite imidazolium framework-67 (ZIF-67) particles to exhibit excellent drag reduction and antifouling performance. This non-smooth surface effectively reduced the drag force by 7.3% compared with the smooth surface, as shown in Figure 13; it also demonstrated significant antifouling effects in contact angle measurements and anti-bioadhesion experiments [109]. A multi-functional biopolymer coating inspired by the skin of the mudskipper significantly reduced the pressure drop in the microfluidic channel by covalently combining chitosan, carboxymethyl cellulose, and methoxypolyethylene glycol amine on the surface of a porous polycaprolactone (PCL) membrane, with a particularly significant drag reduction effect [120]. It can be concluded that the bio-inspired design enhances the drag reduction performance, fouling resistance, and durability of the coating.

The design of the composite coating further optimizes the drag reduction performance. A modified coating composed of epoxy resin (ER) and zinc oxide (ZnO) was shown to significantly reduce fluid drag by forming micro- and nanostructures on the surface of glass and a model ship. Experimental results revealed that the drag reduction efficiency of the coating on the model ship increased from 5% to 21% with an increase in towing force [105]. A low-drag zinc substrate surface prepared using fluorinated polyurethane encapsulation and chemical etching techniques demonstrated excellent drag reduction in microchannels, with the drag reduction rate essentially remaining at 4.0% at different Reynolds numbers [94]. Combining the properties of multiple materials, the composite coating can effectively enhance the drag reduction performance of the coating and maintain stability in different fluid environments.

The durability test results reveal the potential of modified coatings for use in real-world applications. PVDF–PDMS–SiO_2_ multi-stage rough superhydrophobic coatings prepared using the one-step cold spraying method achieved up to 35% maximum drag reduction in the Reynolds number range of 6 × 10^3^ to 4 × 10^4^ [114]. A composite coating based on KH550@SiO_2_ with STA@TiO_2_ maintained a drag reduction efficiency of 27% after 700 h of salt spray corrosion testing, while the contact angle decay was less than 5° [91]. Mechanical stability experiments showed that the maximum contact angle of the SiO_2_/HLR–Si coating, which was cured and treated at 80 °C for 40 min, was 163°, while the sliding angle was only 1.8°. Compared with the 96° pure HLR–Si resin coating, the maximum drag reduction rate of the prepared superhydrophobic coating is 23.4% [104]. Through optimization of the coating formulation and curing process, the durability of the coating can be significantly improved while ensuring drag reduction performance, providing an experimental basis for its long-term application in the fields of marine equipment, pipeline transportation, and underwater vehicles.

Subsequently, future research should focus on the analysis of the drag reduction mechanism under the multi-physical field coupling condition. Current research on biomimetic modified coatings focuses on a single steady state flow field while, in actual engineering applications, coatings need to cope with variable operating conditions, multi-phase flow, and extreme environmental coupling; therefore, both experiments and simulations are important, and comparing the drag reduction rate under the influence of different factors can help to better confirm the performance of the coating, as shown in Figure 14. Under the harsh conditions of sandy water flow, the stability of the air layer and microstructural integrity of the coating surface face greater challenges, and the introduction of antimicrobial, antifouling, and other functions through composite modification can realize the dual effects of drag reduction and surface protection in a single coating [121,122,123]. Antifouling drag reduction coatings based on environmentally friendly polysaccharides exhibit 5% drag reduction and strong lubrication properties with excellent corrosion resistance. The drag reduction mechanism of bio-inspired modified coatings, such as polysaccharides, is shown in Figure 15 [110,111]. In the design of smart coatings, external stimulus response mechanisms that enable the coating to dynamically adjust surface properties in response to changes in the fluid environment may be introduced. This adaptive adjustment capability is expected to result in breakthrough applications in fields such as aerospace and ocean engineering, providing innovative solutions to address energy consumption and environmental pollution.

With continuous progress in the development of materials science and manufacturing technologies, bio-inspired modified coatings have shown broad application prospects in a range of interdisciplinary fields. From ship navigation to microfluidic chip sensors, and from energy conservation to environmental protection, these coatings enable multi-functional integration while reducing fluid resistance through the precise regulation of surface properties, which can be used to improve the performance of microfluidic chip sensors. This provides a new technical path for engineering applications. The experimental data and numerical simulation results indicate that the design of bio-inspired coatings should focus on the uniformity of the microstructure and the stability of the gas layer to achieve a more efficient drag reduction effect and improved performance in practical applications [124,125,126]. These research results lay a solid theoretical foundation for the engineering application of a new generation of bio-inspired drag reduction modified coatings.

## 5. Conclusions

As an intersection of fluid mechanics and materials science, bio-inspired drag reduction technology has recently shown broad application prospects in the energy-saving design of ships and underwater vehicles. Although significant progress has been made in understanding the mechanisms underlying the drag reduction, preparation, and application effect of the three major technical routes of bio-inspired non-smooth surfaces, bio-inspired superhydrophobic surfaces, and bio-inspired modified coatings, certain limitations and bottlenecks remain in each technical route. Although bio-inspired non-smooth surfaces effectively regulate the flow characteristics of the turbulent boundary layer through microstructure design, the complexity of the manufacturing process and the cost of large-area application are still key factors restricting their engineering popularization. The drag reduction effect of non-smooth surfaces is highly dependent on the Reynolds number and flow velocity of the fluid, and its stability in a wide range of fluid conditions still needs to be optimized. Bio-inspired superhydrophobic surfaces significantly reduce fluid resistance due to the slip effect at the gas–liquid interface; however, the stability of the gas layer is insufficient to maintain their function at high Reynolds numbers and in complex fluid environments. Additionally, the durability of the surfaces is poor and susceptible to mechanical abrasion and chemical corrosion, limiting their reliability under long-term service conditions. The bio-inspired modified coating technology results in the synergistic function of drag reduction and antifouling through chemical modification of materials and micro- and nanostructure design; however, the long-term stability and multi-physical field adaptability of the coating need to be improved, especially under extreme working conditions such as sandy water flow and multi-phase flow. Furthermore, the microstructural integrity and functional stability of the coating surface face greater challenges.

In the future, research should prioritize clarifying drag reduction mechanisms and optimizing material properties under multi-physical field coupling conditions, with a focus on establishing quantitative correlations between these mechanisms, material properties, and actual performance. For bio-inspired non-smooth surfaces, developing efficient, low-cost manufacturing processes remains critical for advancing their engineering applications. Specifically, synergistic innovation in additive manufacturing and micro- and nanoprocessing technologies should be leveraged to achieve the large-area, high-precision fabrication of complex microstructures while reducing production costs—efforts that should be validated through systematic experimental studies. Additionally, exploring the quantitative relationships between microstructure parameters and hydrodynamic performance, as well as optimizing the drag reduction efficacy of non-smooth surfaces across a broad range of Reynolds numbers, is expected to provide a foundational basis for their widespread application in shipping, aerospace, and other fields.

For bio-inspired superhydrophobic surfaces, research should concentrate on enhancing gas layer stability and surface durability, with an emphasis on quantifying the interplay between these two properties and drag reduction performance. This can be achieved by constructing dynamic self-repairing coatings and intelligent responsive materials, enabling active maintenance of the gas layer and adaptive adjustment of surface functions during long-term service. Simultaneously, integrating bio-inspired design principles with cutting-edge materials science technologies to develop composite systems with both high hydrophobicity and strong mechanical stability is expected to significantly improve the reliability of superhydrophobic surfaces in complex fluid environments, particularly if supported by rigorous performance testing.

The development of bio-inspired modified coating technology requires deeper multi-disciplinary integration, particularly exploring synergistic innovation pathways across material chemistry, fluid mechanics, and intelligent manufacturing. Through the incorporation of multi-functional materials and intelligent coating designs, modified coatings can achieve comprehensive performance improvements in drag reduction, antifouling, and anticorrosion—with clear quantitative metrics to evaluate these enhancements. Under complex working conditions, optimizing the micro- and nanostructural designs and chemical composition of coatings is expected to notably enhance their stability and adaptability under multi-physical field coupling, necessitating systematic validation through combined experimental and simulation approaches. Furthermore, the synergistic advancement of computational fluid dynamics and experimental technologies will deepen understanding of the quantitative correlations between microstructural designs and macroscopic fluid behaviors, ultimately providing robust theoretical and experimental support for the engineering application of next-generation high-efficiency drag-reducing coatings.

Bio-inspired drag reduction technology, as an important means to solve the energy crisis and environmental problems, has broad research and application prospects. Focusing on the technical bottlenecks and future development directions, and promoting the synergistic innovation of non-smooth surfaces, superhydrophobic surfaces, and modified coating technologies are expected to generate more efficient and reliable technical solutions for the energy-saving design of ships and underwater vehicles, while providing new impetus for the sustainable development of related fields.

## Figures and Tables

**Figure 1 biomimetics-10-00470-f001:**
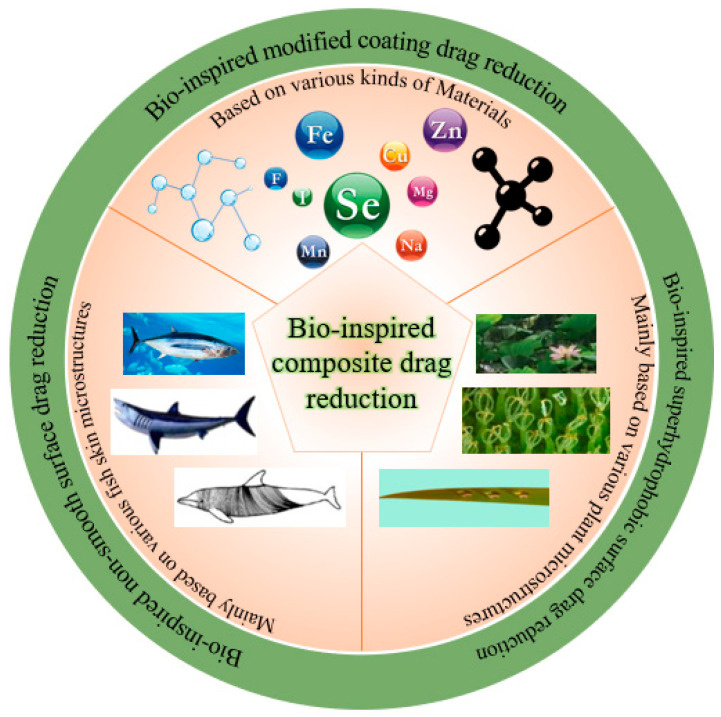
Various bio-inspired drag reduction methods.

**Figure 2 biomimetics-10-00470-f002:**
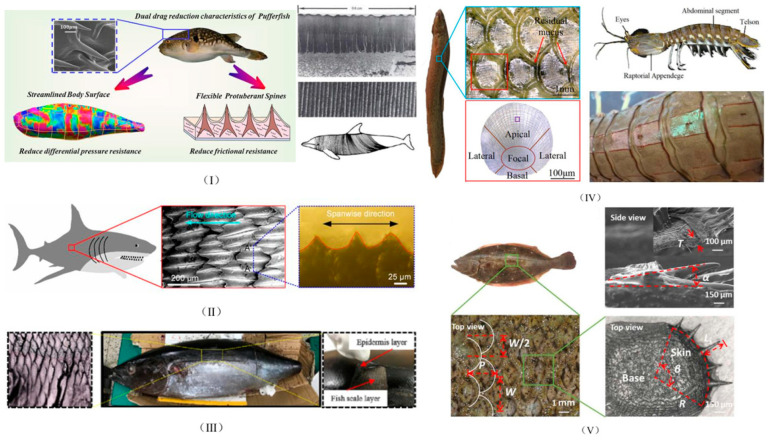
Surface microstructural features of mainstream bio-inspired objects: (**I**) streamline d body surface of pufferfish and morphology of dolphin skin [1,2]. (**II**) Shark scale structure and SEM images of shark skin [23]. (**III**) Multi-layered structure of tuna skin and its morphological features [26]. (**IV**) Surface microstructural morphology of mudskipper and dorsal and lateral views of the abdominal segments of mantis shrimp [30,34]. (**V**) Flounder skin scales and structural characteristics of individual scales [4].

**Figure 3 biomimetics-10-00470-f003:**
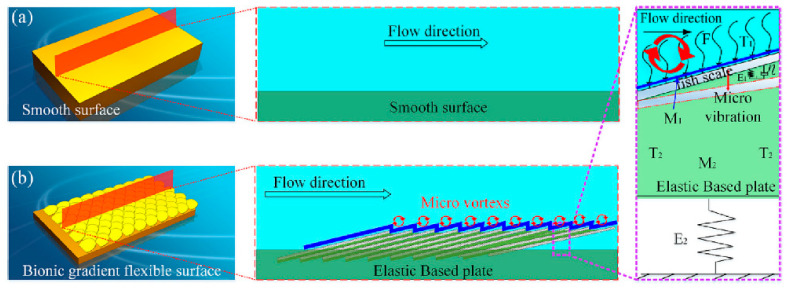
The schematic diagram of the drag reduction mechanism. (**a**) Smooth surface; (**b**) bio-inspired gradient flexible non-smooth surface [12].

**Figure 5 biomimetics-10-00470-f005:**
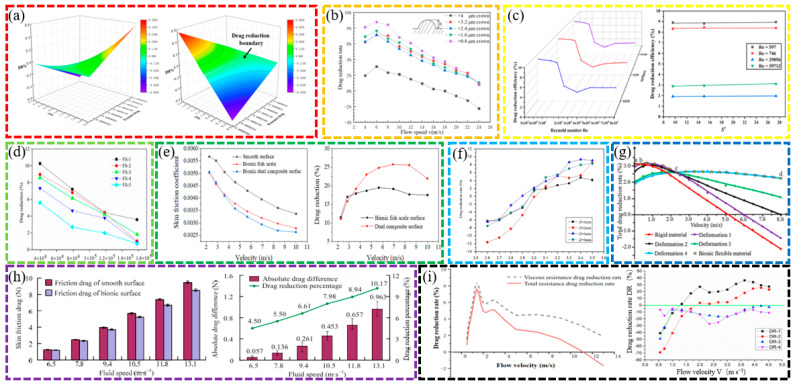
Effect of various factors on the rate of drag reduction [1,3,10,11,17,21,26,28,30,31]. (**a**) Spatial curve diagram of FPr, pressure drag and drag reduction rate. (**b**) Relationship between the water flow velocity and the drag reduction rate of the groove surface of the positive crown with different heights. (**c**) Drag reduction report according to the inflow velocity and dimensionless spacing. (**d**) Total drag reduction rate for biomimetic fish scale at different Re numbers. (**e**) Skin friction coefficient and drag reduction rate. (**f**) Curves of drag reduction rate for different jet diameters on bio-inspired surface. (**g**) Total drag reduction rate under different deformations. (**h**) The drag reduction efficiency of bio-inspired surface in variation with flow speed. (**i**) The relationship between drag reduction rate and flow velocity.

**Figure 7 biomimetics-10-00470-f007:**
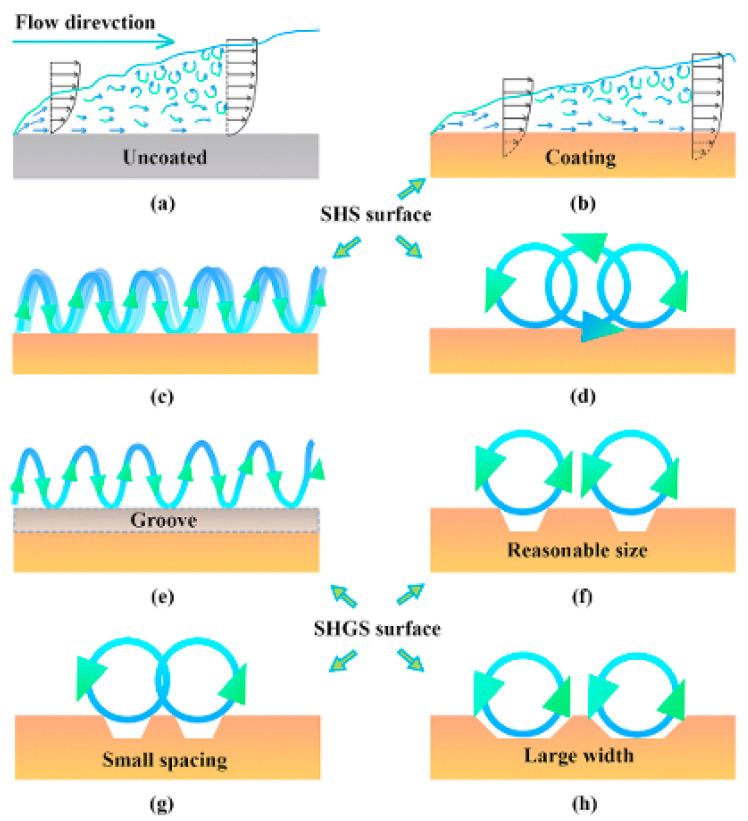
Schematic diagram of the drag reduction principles for SHS and SHGS surfaces [49].

**Figure 8 biomimetics-10-00470-f008:**
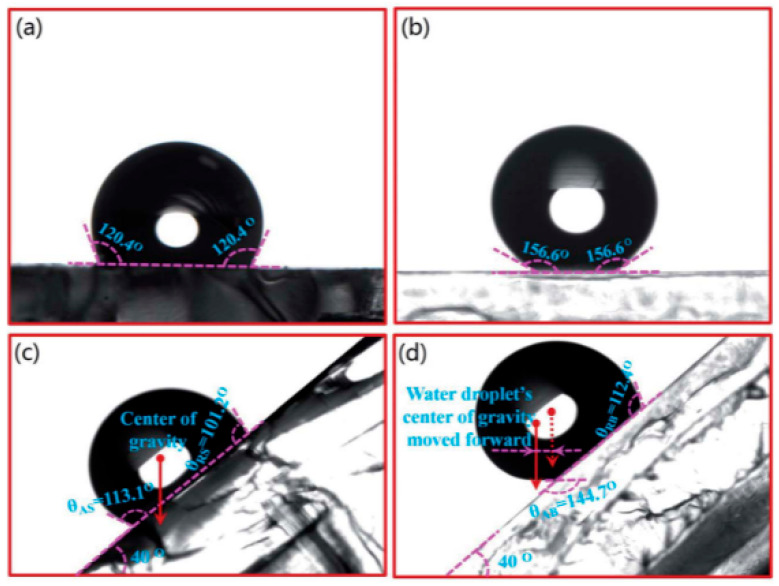
The wettability of smooth surface and bio-inspired superhydrophobic surface. (**a**) Static contact angle of smooth surface. (**b**) Static contact angle of bio-inspired superhydrophobic surface. (**c**) Rolling state of droplet on a smooth surface. (**d**) Rolling state of droplet on the bio-inspired superhydrophobic surface [56].

**Figure 9 biomimetics-10-00470-f009:**
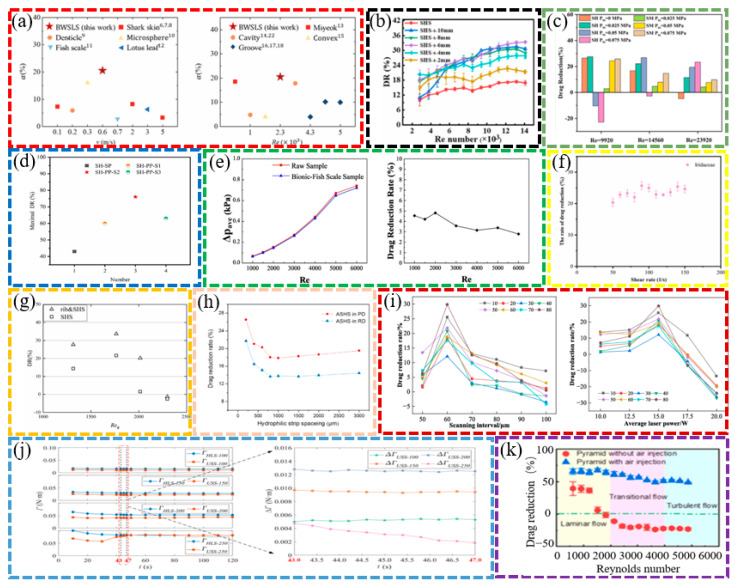
Drag reduction effect of bio-inspired superhydrophobic surfaces under different Reynolds numbers and other conditions [37,38,48,51,60,62,65,67,68,71,73]. (**a**) Drag reduction rates of the BWSLS and other biomimetic and single structured surfaces obtained at different speeds and Reynolds numbers. (**b**) DR for surfaces with different superhydrophobic strips at different Reynolds numbers. (**c**) Drag reduction rate of different conditions. (**d**) Maximum drag reduction rate of samples with different apertures. (**e**) Comparison of pressure difference and drag reduction at different Reynolds numbers. (**f**) Relationship between shear rate and drag reduction rate on the Iridaceae leaf. (**g**) Drag reduction efficiency of rib and SHS and SHS at different Reynolds numbers. (**h**) Drag reduction ratios of the as-prepared ASHS sample along the PD and RD with different hydrophilic strip spacings at the same velocity. (**i**) Drag reduction rate for the untreated microchannel and LACE-treated microchannels with different scan intervals and average laser power at different Reynolds numbers. (**j**) Drag reduction effect of USS covered with air layer compared with HLS without air layer at different rotation speeds. (**k**) Calculated drag reduction of the pyramid-shaped superhydrophobic surfaces with and without air injection as a function of the Reynolds number.

**Figure 10 biomimetics-10-00470-f010:**
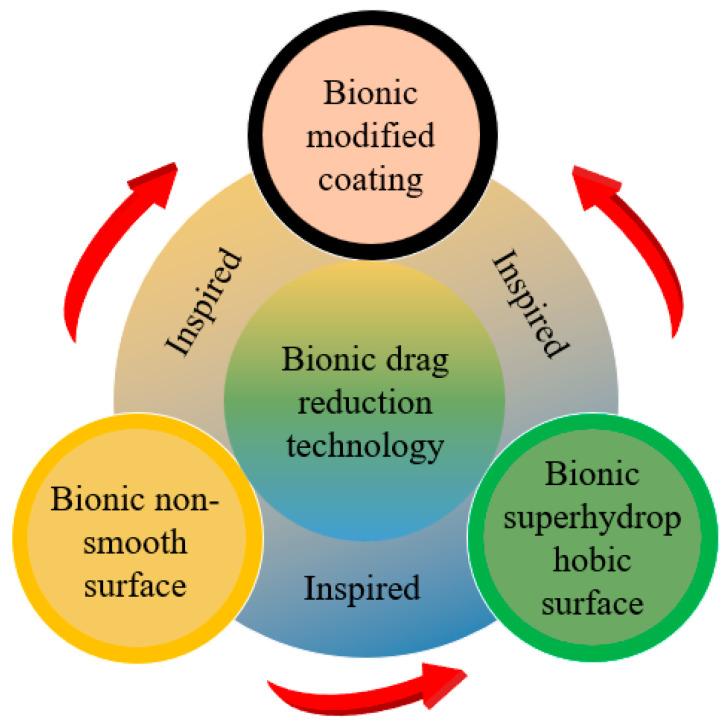
Emerging modified coating technologies inspired by the characteristics of fundamentally non-smooth and superhydrophobic surfaces.

**Figure 11 biomimetics-10-00470-f011:**
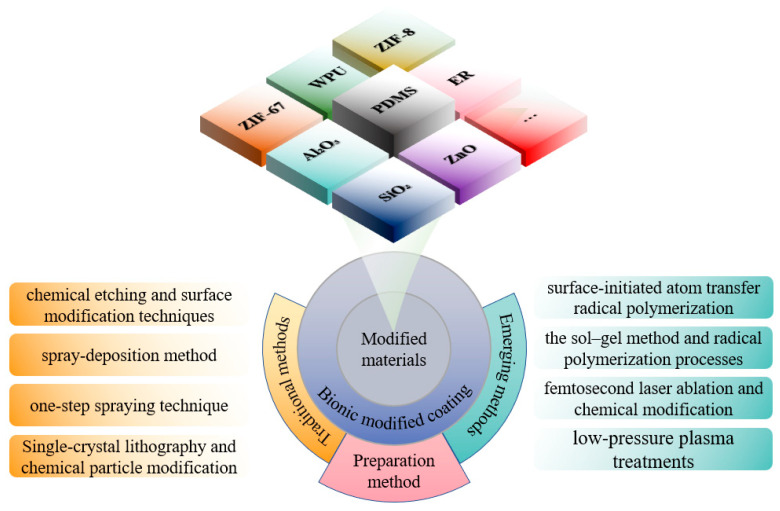
Summary of modified materials and coating fabrication methods.

**Figure 12 biomimetics-10-00470-f012:**
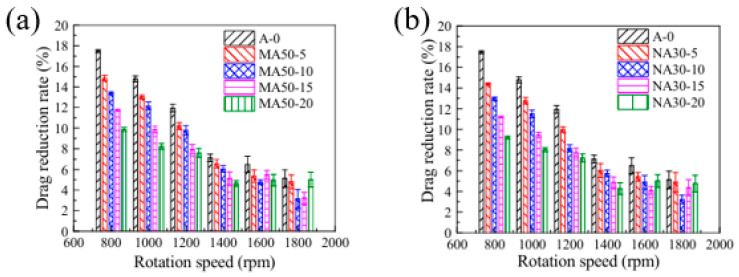
Drag reduction rate of coatings: micro-alumina (**a**) and nano-alumina (**b**) [101].

**Figure 13 biomimetics-10-00470-f013:**
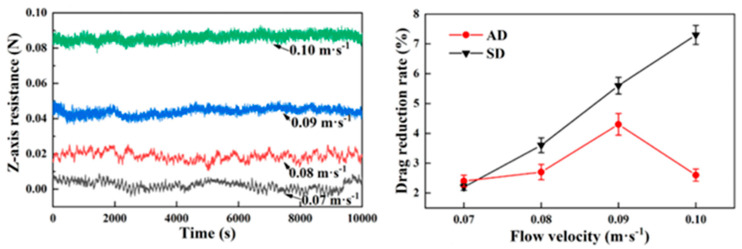
Drag of smooth measuring surface and drag reduction for non-smooth surfaces [109].

**Figure 14 biomimetics-10-00470-f014:**
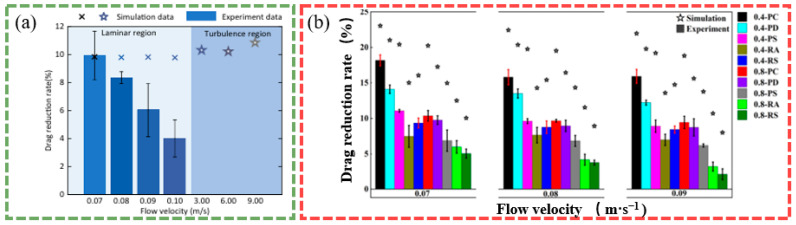
Comparison of experimental and simulated drag reduction rates for (**a**) different flow rates [86] and (**b**) different printed surfaces [88].

**Figure 15 biomimetics-10-00470-f015:**
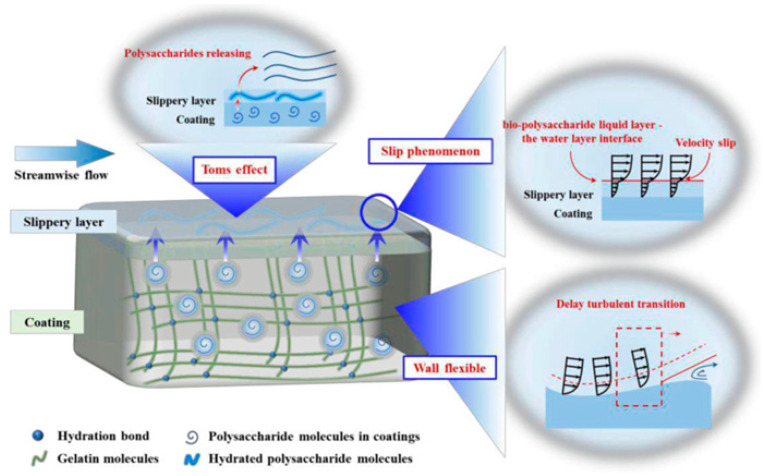
Drag reduction mechanisms of the gelatin–polysaccharide coating [110].

**Table 1 biomimetics-10-00470-t001:** Summary of non-smooth surface manufacturing methods.

Investigators	Method	Schematic Diagram	Drag Reduction Rate
Xiaoming Feng et al. [13]	Combining the sintering and coating processes	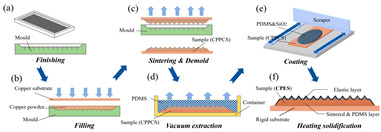	11.5–17.5%
Tao Wu et al. [11]	Micro-end-milling and micro-fly-milling	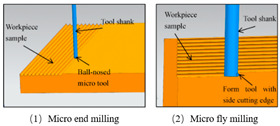	For fluid velocities of 0.5–4.5 m/s, the average drag reduction rate is 13.05%.
Yang Zhang et al. [22]	Shaft-to-shaft roller shaft hot embossing process	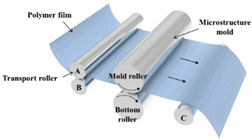	The curved groove reduces drag by 10.3%, while the straight groove reduces it by 8%
Liyan Wu et al. [31]	computer numerical control (CNC) machining	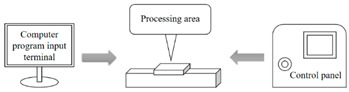	Reaches 23% at a speed of 1.683 m/s
Dengke Chen et al. [12]	The three-dimensional printing method, spraying, and casting process	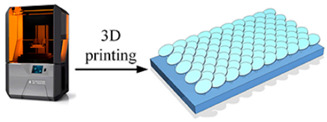	The Reynolds (Re) number was 6.8 × 10^4^; the maximum drag reduction ratio could reach 13.8%
Dengke Chen et al. [28]	Laser ablating and casting process	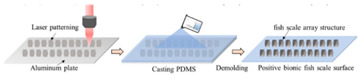	The maximum drag reduction was 10.26% at a Reynolds number of 39,532

**Table 2 biomimetics-10-00470-t002:** Summary of superhydrophobic surface manufacturing methods.

Investigators	Method	Schematic Diagram	CA/Drag Reduction Rate
Yang Liu et al. [59]	Projection micro-stereolithography (PlSL) 3D printing technology	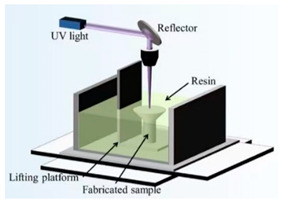	160°/—
Yangmin Chen et al. [60]	3D printing selective laser melting technique with silanized silica nanoparticle spraying		Average contact angle of 158.4°/maxima 20.56%
Jiangpeng Qu et al. [49]	Laser etching and spray coating	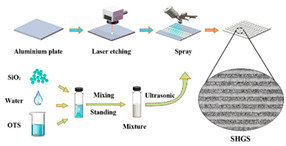	—/Improvement of 7% to 26.51% compared to the superhydrophobic coated surface without grooves
Wanting Rong et al. [37]	Laser ablation and surface fluorination	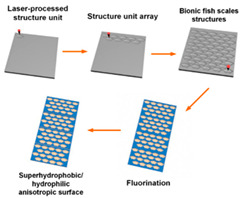	—/As the velocity increases, the ASHS can be held near 40%.
Huan Wang et al. [56]	The combination method of photolithography and vacuum casting	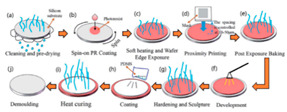	—/Maximum 6.29%
Changzhuang Yao et al. [61]	Wet etching	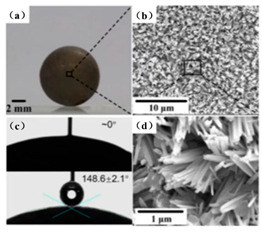	148.6 ± 2.1°/The sphere with a cavity can reduce the hydrodynamic drag by more than 90%, compared with the bare spheres
Liangpei Zhang et al. [74]	Electrostatic flocking and subsequent surface modification	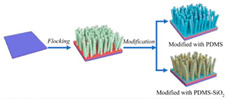	—/The DR value is 21% and 28% for the Flock@PDMS and Flock@PDMS-SiO2
Zhe Li et al. [66]	Electrochemical etching and anodization	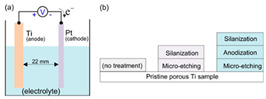	—/92–96% for high-speed water jetting

## Data Availability

No new data were created or analyzed in this study. Data sharing is not applicable to this article.

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
