# Peer review of "Underwater Drag Reduction Applications and Fabrication of Bio-Inspired Surfaces: A Review"

_biomimetics, 2025, doi:10.3390/biomimetics10070470_

Round 1
Reviewer 1 Report
Comments and Suggestions for Authors
The review manuscript by Z. Zhang et al. provides an overview on and review of underwater drag reduction applications and fabrication methods, that have been bio-inspired by biological surfaces. The authors follow the categorization into three majpr technological paths, i.e. no-smotth surfaces, superhydrophobic surfaces, as well as modified coatings. After shortly introducing the biological models, for each of these paths the various fabrication methods and drag reduction effects are listed and described in separte chapters, followed by a critical outlook and conclusion.
The manuscript gives an interesting overview summary of the current status-quo and cites more than 100 refs. However, a few aspects need refinement:
1) Since the focus of the review is the field of underwater drag reduction, the reviewer proposes that the title should reflect this aspect more clearly.
2) The three major technological paths leaves out the phenomenon of supercavitation, which is also an important aspect of bio-inspired drag reduction.The authors are asked to comment on this, or also list this aspect in the review.
3) Non-smooth surfaces also list the dolfin skin, but the explanation leaves out the effect of the visco-elastic properties of the fat layer underlying the skin. This has been reported to generate ridge deformations of the skin at over-critical swimming speed perpendicular to the body axis/flow direction, causing bond vortexes that help to delay turbulent transitions and increased boundary layer formation.
4) In section 2.3 p.9, the "disordered flow field near the wall" should be visible in Fig. 3A, however, the schematic diagram does not show a disordered flow field. Thus, it remains unclear here what the authors want to explain.
5) In all sections, but particularly in section 3, the original literature of the bio-inspiring models are missing. Either it remains elusive from which source the authors gathered the informations, or often non-original, more recent refs have been cited.
6) Concerning refs, the manuscript is not properly prepared and edited, since it contains numerous ref source errors. The authors are asked to correct this issue in all instances.
7) The individual parts of Figs. 4, 5, and 9 are way too small, and readability is impaired. The authors are asked to re-organize these Figs. for clearer visiblity.
Comments on the Quality of English LanguageThe quality of English language needs to be impvoved, especially the grammar in some sentences.
Author Response
1) Since the focus of the review is the field of underwater drag reduction, the reviewer proposes that the title should reflect this aspect more clearly.
Response: We agree with the reviewer's suggestion. The title has been revised to explicitly reflect the focus on underwater drag reduction, ensuring clarity on the review's core scope. The new title has been changed to “Underwater drag reduction applications and fabrication of bio-inspired surfaces: A review”.
2) The three major technological paths leaves out the phenomenon of supercavitation, which is also an important aspect of bio-inspired drag reduction.The authors are asked to comment on this, or also list this aspect in the review.
Response: Thank you for pointing out this oversight. Supercavitation, as an important bio-inspired drag reduction phenomenon, has been added as a supplementary technological path with relevant elaboration in the review.
In section 3.3 p.20: " In addition, as an important phenomenon of bionic drag reduction, the core principle of supercavitation is to transform the solid–liquid interface into a solid–air interface through the surface gas film to reduce drag. There is a common interface control logic with the drag reduction mechanism underlying superhydrophobic surfaces. The ability of the superhydrophobic micro- and nanostructures to capture bubbles can assist in stabilizing the supercavitating gas film, and the gas film formed via supercavitation can enhance the drag reduction effect of the superhydrophobic surface. At the technical level, the combination of supercavitation and the superhydrophobic surface has become a research hotspot: the low surface energy characteristics of superhydrophobic surfaces can reduce the attachment threshold of cavitation bubbles and promote the formation of gas films; at the same time, the low surface energy characteristics of superhydrophobic surfaces can reduce the adhesion threshold of cavitation bubbles and promote the formation of gas films. The micro-structure can anchor the boundary of the gas film, inhibit the collapse of the cavitation bubble, and reduce the instability problem associated with traditional supercavitation. This synergistic system not only retains the stable drag reduction advantage of the superhydrophobic surface in medium and low speed scenarios, but also overcomes the drag reduction bottleneck at high flow rates through the cavitation effect. An important direction for superhydrophobic surface drag reduction technology involves expanding to include more diverse working conditions. "
3) Non-smooth surfaces also list the dolfin skin, but the explanation leaves out the effect of the visco-elastic properties of the fat layer underlying the skin. This has been reported to generate ridge deformations of the skin at over-critical swimming speed perpendicular to the body axis/flow direction, causing bond vortexes that help to delay turbulent transitions and increased boundary layer formation.
Response: We appreciate the reviewer's valuable insight. The effect of the visco-elastic properties of the fat layer under dolphin skin, including its role in ridge deformation and turbulent transition delay, has been supplemented in the corresponding section.
In section 2.1 p.4: " The dolphin's flexible skin reduces drag through elastic vibration absorption, and is structured with transversely corrugated microgrooves that are capable of significantly affecting the underwater drag reduction rate by altering the flow velocity and drag distribution near the wall and increasing the thickness of the viscous bottom layer [1]. The skin of dolphins, along with the visco-elastic properties of the underlying fat layer, has been reported to produce ridge-like deformations at supercritical swimming speeds that are perpendicular to the body axis and flow direction. These deformations generate bound vortices, which help to delay the transition to turbulence and enhance boundary layer formation. "
4) In section 2.3 p.9, the "disordered flow field near the wall" should be visible in Fig. 3A, however, the schematic diagram does not show a disordered flow field. Thus, it remains unclear here what the authors want to explain.
Response: We acknowledge the issue. Additional explanatory notes have been added to Fig. 3A to provide a clear understanding of the nearby flow field.
In section 2.3 p.9: " Smooth surfaces struggle to effectively handle complex flow behaviors near the wall in turbulent environments, as illustrated in Fig. 3(a), while bio-inspired non-smooth surfaces show significant drag reduction advantages by virtue of their unique structure, as shown in Fig. 3(b). "
5) In all sections, but particularly in section 3, the original literature of the bio-inspiring models are missing. Either it remains elusive from which source the authors gathered the informations, or often non-original, more recent refs have been cited.
Response: Thank you for the reminder. Original literature on bio-inspiring models has been supplemented in all relevant sections, and citations have been adjusted to prioritize foundational studies where appropriate.
6) Concerning refs, the manuscript is not properly prepared and edited, since it contains numerous ref source errors. The authors are asked to correct this issue in all instances.
Response: We apologize for the citation errors. All reference sources have been thoroughly checked and corrected to ensure accuracy and proper formatting.
7) The individual parts of Figs. 4, 5, and 9 are way too small, and readability is impaired. The authors are asked to re-organize these Figs. for clearer visiblity.
Response: We agree with the comment. Figs. 4, 5, and 9 have been reorganized, with individual parts enlarged and layout optimized to improve readability.
8) The quality of English language needs to be impvoved, especially the grammar in some sentences.
Response: We appreciate the reviewer's reminder regarding English language quality. The manuscript has undergone professional English editing by MDPI, with grammar and technical terms carefully checked and revised to meet scholarly journal standards. We believe this enhances the clarity and accuracy of the text.
Reviewer 2 Report
Comments and Suggestions for Authors
The article systematically compiles the latest research progress of bio-inspired micro-structured surfaces in the field of underwater drag reduction, and focuses on analyzing the drag reduction mechanism, preparation process and application effect of the three major technology paths, namely, bio-inspired non-smooth surfaces, bio-inspired super-hydrophobic surfaces and bio-inspired modified coatings. The article is clear, rich in content and illustrated, and has important theoretical significance and engineering application value for the energy-saving design of ships and underwater vehicles, especially in the analysis of drag reduction mechanism and optimization of material properties under the condition of multi-physical field coupling, which shows a certain degree of innovation and foresight. There is still room for further improvement in the following aspects:
- The research background and importance of the bio-inspired drag reduction technology are described in the introduction, but it is suggested that the authors further compare the advantages and limitations of the three major technology paths in different application scenarios, as well as the prospective outlook of the future research direction, in order to enhance the scientific contribution of the article.
- The current chapter arrangement of the article is reasonable, but between the chapters, it is recommended that the authors further optimize the transition statements to make the logic of the article more coherent and enhance the logic.
- The article contains a large number of charts and graphs, which are informative and have visualization value. However, some of the diagrams (e.g., Figure 1, etc.) are not described in sufficient detail, and it is recommended that specific image descriptions, etc., be added so that readers can understand the information shown in the diagrams without having to consult the main text. In addition, it is recommended to check whether all the diagrams are clearly guided when they are first mentioned in the text.
- The overall language of the paper is relatively smooth, but there are some expression problems in some statements, for example:
“bio-inspired” and “bionic” are mixed, and it is recommended to unify them into “bio-inspired”;
‘SHS’ and ”SHGS" were not defined when they first appeared, and it is suggested that the authors check the article for unreasonable expressions and then correct them.
- In the research outlook of the “Conclusion” section, it is suggested that the authors further specify and refine the description of future research directions, and list some specific technical approaches or research ideas, such as exploring new micro-nano-processing technologies, optimizing the existing manufacturing process parameters, etc. By specifying the outlook of the research, a clearer definition of the subsequent research work can be provided. subsequent research work to provide more clear guidance and reference.
- The cited literature in this paper is large and covers a wide range, and it is suggested that the authors add some latest research results published in recent years in the relevant chapters such as biomimetic superhydrophobic surfaces (eg. Advanced Functional Materials. 2024, 2413552; Materials Science & Engineering R: Reports. 2024, 161: 100862.). This will help to make the review content of the paper more cutting-edge and comprehensive, so that readers can understand the latest research dynamics and progress in the field.
Author Response
- The research background and importance of the bio-inspired drag reduction technology are described in the introduction, but it is suggested that the authors further compare the advantages and limitations of the three major technology paths in different application scenarios, as well as the prospective outlook of the future research direction, in order to enhance the scientific contribution of the article.
Response: We appreciate the suggestion. We have added a comparative analysis of the advantages, limitations, and application scenarios of the three major technology paths, and refined the prospective outlook on future research directions to enhance the scientific contribution.
In section 1 p.3: " Although the mechanisms of the three technical approaches are different, they all follow the principle of synergy between structure and function and complement each other in key aspects of fluid mechanics, such as reducing turbulence fluctuation and delaying flow separation. In practical applications, bio-inspired non-smooth surfaces adapt to the multiphase flow environment, but their widespread adoption is limited by their processing cost; bio-inspired superhydrophobic surfaces provide significant drag reduction in microfluidics and other scenarios, but their durability is insufficient; modified coating surfaces have a wide range of applications, but are greatly influenced by environmental factors. In the future, composite applications can be explored to take performance into account, intelligent adaptive surfaces can be developed to deal with variable flow fields, and the correlations between microscopic mechanisms and macroscopic effects can be deepened to further enhance the practical value and scientific contribution of the technology. "
- The current chapter arrangement of the article is reasonable, but between the chapters, it is recommended that the authors further optimize the transition statements to make the logic of the article more coherent and enhance the logic.
Response: Thank you for the comment. We have optimized the transition statements between chapters to improve the logical coherence of the manuscript.
In section 4.1 p.20: " On the basis of exploring bio-inspired non-smooth surfaces and bio-inspired superhydrophobic surfaces, remarkable progress has been made in integrating these two types of surfaces into composite superhydrophobic grooved surfaces, resulting in a promising approach for the development of bio-inspired superhydrophobic surfaces. The synergistic advantages of these two structures can be fully utilized. However, despite the significant potential of bio-inspired superhydrophobic surfaces in the field of drag reduction, challenges remain in terms of the stability and durability of the gas layer, especially in complex fluid environments. Through continuous research on the characteristics of non-smooth and superhydrophobic surfaces and to further enhance the drag reduction performance and realize the multifunctional synergistic effect, researchers have begun to explore the emerging biomimetic modified coating technology based on the design of micro- and nanostructures and chemical modification of the materials, as shown in Fig. 10. "
- The article contains a large number of charts and graphs, which are informative and have visualization value. However, some of the diagrams (e.g., Figure 1, etc.) are not described in sufficient detail, and it is recommended that specific image descriptions, etc., be added so that readers can understand the information shown in the diagrams without having to consult the main text. In addition, it is recommended to check whether all the diagrams are clearly guided when they are first mentioned in the text.
Response: We agree with the reviewer's point. Figure 1 has been modified to ensure that readers can understand the information on their own. Additionally, we have checked and ensured all diagrams are clearly referenced when first mentioned.
- The overall language of the paper is relatively smooth, but there are some expression problems in some statements, for example:
“bio-inspired” and “bionic” are mixed, and it is recommended to unify them into “bio-inspired”;
‘SHS’ and ”SHGS" were not defined when they first appeared, and it is suggested that the authors check the article for unreasonable expressions and then correct them.
Response: We apologize for the inconsistent expressions. "Bionic" has been unified as "bio-inspired", and definitions for "SHS" and "SHGS" have been added when they first appear. Other unreasonable expressions have also been corrected.
In section 3.3 p.17: " Superhydrophobic surfaces (SHSs) show remarkable potential for application in drag reduction due to their unique micro- and nanostructures and low surface energy properties. In addition, the unique properties of the superhydrophobic grooved surfaces (SHGSs) resulting from the grooved structure give the surfaces superhydrophobic properties and can also regulate the behavior of the flow of liquids through the geometric parameters of the groove, which has important research value in micro-structure flow field analysis. In recent years, researchers have investigated the drag reduction performance of superhydrophobic surfaces under different fluid environments and Reynolds numbers via various experimental and simulation methods. The drag reduction principles of SHS and SHGS are shown in Fig. 7. "
- In the research outlook of the “Conclusion” section, it is suggested that the authors further specify and refine the description of future research directions, and list some specific technical approaches or research ideas, such as exploring new micro-nano-processing technologies, optimizing the existing manufacturing process parameters, etc. By specifying the outlook of the research, a clearer definition of the subsequent research work can be provided. subsequent research work to provide more clear guidance and reference.
Response: Following the suggestion, the research outlook in the "Conclusion" section has been specified and refined, including specific technical approaches such as exploring new micro-nano processing technologies and optimizing manufacturing parameters, to provide clearer guidance for subsequent research.
- The cited literature in this paper is large and covers a wide range, and it is suggested that the authors add some latest research results published in recent years in the relevant chapters such as biomimetic superhydrophobic surfaces (eg. Advanced Functional Materials. 2024, 2413552;Materials Science & Engineering R: Reports. 2024, 161: 100862.). This will help to make the review content of the paper more cutting-edge and comprehensive, so that readers can understand the latest research dynamics and progress in the field.
Response: We appreciate the reviewer's valuable suggestion. We include these latest research papers (Advanced Functional Materials. 2024, 2413552; Materials Science & Engineering R: Reports. 2024, 161: 100862.) in the section on cutting-edge applications. The potential applications of these bio-inspired high-performance flexible sensors in underwater scenarios enrich the elaboration on cutting-edge developments in the field, making the review content more cutting-edge and comprehensive, and helping readers understand the latest progress in the field.